# Differential Effects of Dietary Patterns on Advanced Glycation end Products: A Randomized Crossover Study

**DOI:** 10.3390/nu12061767

**Published:** 2020-06-12

**Authors:** Yoona Kim, Jennifer B. Keogh, Permal Deo, Peter M. Clifton

**Affiliations:** 1Department of Food and Nutrition, Institute of Agriculture and Life Science, Gyeongsang National University, Jinju 52828, Korea; yoona.kim@gnu.ac.kr; 2Health and Biomedical Innovation, Clinical and Health Sciences, University of South Australia, Adelaide SA 5000, Australia; Jennifer.Keogh@unisa.edu.au (J.B.K.); Permal.Deo@unisa.edu.au (P.D.)

**Keywords:** dietary advanced glycation products, carboxymethyl–lysine (CML), carboxyethyl–lysine (CEL), methylglyoxal–hydroimidazalone (MG-H1)

## Abstract

Dietary advanced glycation end products (AGEs) are believed to contribute to pathogenesis of diabetes and cardiovascular disease. The objective of this study was to determine if a diet high in red and processed meat and refined grains (HMD) would elevate plasma concentrations of protein-bound AGEs compared with an energy-matched diet high in whole grain, dairy, nuts and legumes (HWD). We conducted a randomized crossover trial with two 4-week weight-stable dietary interventions in 51 participants without type 2 diabetes (15 men and 36 women aged 35.1 ± 15.6 y; body mass index (BMI), 27.7 ± 6.9 kg/m^2^). Plasma concentrations of protein-bound Nε-(carboxymethyl) lysine (CML), Nε-(1-carboxyethyl) lysine (CEL) and Nδ-(5-hydro-5-methyl-4-imidazolon-2-yl)-ornithine (MG-H1) were measured by liquid chromatography–tandem mass spectrometry (LC-MS/MS). The HMD significantly increased plasma concentrations (nmol/mL) of CEL (1.367, 0.78 vs. 1.096, 0.65; *p* < 0.01; *n* = 48) compared with the HWD. No differences in CML and MG-H1 between HMD and HWD were observed. HMD increased plasma CEL concentrations compared with HWD in individuals without type 2 diabetes.

## 1. Introduction

The accumulation of advanced glycation end products (AGEs) is associated with accelerated aging [1] and may be involved in the development of degenerative disease––such as diabetes, cardiovascular disease and Alzheimer’s disease––by promoting insulin resistance, inflammation and oxidative stress [2,3,4,5,6,7].

AGEs can be endogenously generated and also enter the body when AGEs are consumed from food [8]. Approximately 10–30% of consumed AGEs are absorbed in the intestine. About 1/3 of consumed AGEs are excreted via urine or feces and 2/3 accumulate in the body [9]. AGE formation occurs through nonenzymatic reactions among free carbonyl groups of reducing sugars and reactive aldehydes derived from the free amino groups (lysine or arginine) in proteins, lipids and nucleic acids from rearrangement of the Schiff base and Amadori products, known as the Maillard reaction [8,10]. AGEs are categorized into fluorescent cross-linking AGEs (pentosidine and crossline), non-fluorescent cross-linking AGEs (imidazolium dilysine cross-links, alkyl formyl glycosyl pyrrole (AFGP) cross-links and arginine-lysine imidazole (ALI) cross-links), and non-cross-linking AGEs (pyrraline, carboxymethyllysine (CML) and carboxyethyllysine (CEL)) [11]. CML, an indicator of Maillard reaction product, is a major AGE compound [11]. CML and CEL are lysine modificationwhereas methylglyoxal hydroimidazolone (MG-H1—mainly generated through highly reactive α-dicarbonyl, e.g., methylglyoxal) is an arginine adduct [12].

Higher levels of AGEs are produced in thermally processed foods under dry conditions, such as by grilling, broiling, frying and roasting, compared with foods processed slowly at lower temperatures or in water [13]. Foods contain high levels of AGEs when they are exposed to short-time processing with dry heat (e.g. baking, grilling, barbecuing, broiling, searing, frying, toasting and roasting), high temperature and increased pH compared with longer-time processing with lower temperature and water (e.g. boiling and steaming) [14].

The formation of AGEs can potentially be reduced when red meat is consumed with abundant unprocessed plant foods (spices, herbs, fruits and vegetables) [15,16,17,18,19,20]. It is uncertain whether dietary AGEs can play a role in the etiologies of type 2 diabetes mellitus (T2DM) and cardiovascular disease (CVD) or whether only endogenously generated AGEs contribute to these diseases. However, high red meat consumption (especially processed meat) as part of a low-antioxidant, low-fiber, highly available starch standard western diet can increase dietary AGEs [21]. The effect of dietary AGEs on disease risk markers can differ according to a person’s health status (healthy, with or without T2DM or with CVD) [2,22,23].

A meta-analysis of 17 randomized controlled trials (RCTs), comparing low-AGE diets with high-AGE diets indicated that low-AGE diets could reduce risk markers of cardiometabolic disease, such as insulin, LDL cholesterol, CRP and adhesion molecules, and inflammatory markers [22]. A second, smaller meta-analysis found similar effects and noted a high-AGE diet increased circulating AGEs in most, but not all, studies (as assessed using ELISA) [2].

As we previously reported [24,25], a diet high in red and processed meat and refined grains (HMD) significantly attenuated the insulin sensitivity index (ISI) and significantly increased insulin and glucose levels only in relatively insulin-resistant subjects (fasting insulin > 56 pmol/L, *n* = 25), but this was not the case in insulin-sensitive subjects (insulin < 56 pmol/L, *n* = 24) compared with a diet high in whole grain, nuts, dairy and legumes (HWD). No differences in hs-CRP, IL-6, fluorescent AGEs or plasma CML (measured by ELISA) were observed between the HMD and HWD [25].

To date, no interventions have examined the effect of different dietary patterns on plasma concentrations of advanced glycation end products measured by LC-MS/MS. Only one study has shown increased urinary AGEs measured by LC-MS/MS with a high-AGE diet versus a low-AGE diet [26]. We have already measured CML by immunoassay and fluorescent AGEs in our previous publication [25]. However, the method of liquid chromatography–tandem mass spectrometry (LC-MS/MS) may be a more reliable method to study AGEs [27,28].

This present study aimed to investigate the association between dietary patterns and protein-bound AGEs (CML, CEL and MG-H1) using LC-MS/MS, which is a far more specific and sensitive method as the immunoassay may suffer from steric hindrance and interference from anti-AGE antibodies [27,28]. It also aimed to examine the association between plasma advanced glycation end products (CML, CEL and MG-H1) and the changes in ISI and other biomarkers found in previously published studies [24,25].

We hypothesized that subjects without T2DM on the HMD would have higher plasma concentrations of advanced glycation end products compared with those on the HWD.

## 2. Materials and Methods

### 2.1. Ethical Approval and Registration 

The protocol was approved by the University of South Australia Human Research Ethics committee (0000032778), and all participants provided their written informed consent prior to commencement. This trial was registered at the Australian New Zealand Clinical Trials Registry (ACTRN12614000519651).

### 2.2. Study Participants

A total of 51 subjects without T2DM (body mass index (BMI) 18–45 kg/m^2^) and aged > 18 y were recruited as described in detail elsewhere [16,17,19]. Medication or supplements influencing glucose metabolism, food allergy or lactose intolerance, a history of metabolic illness—such as liver or kidney disease—pregnancy or breastfeeding, significant weight gain or weight loss (±3 kg) over the previous 3 months were exclusion criteria. In older people and people with T2DM, the levels of protein-bound AGEs are high and it would be difficult to see any influence of short-term dietary changes on protein-bound AGEs. Thus, we hoped with a population with a mean age of 35 years we would be more likely to see dietary influences. The only previous study examining dietary AGEs and insulin sensitivity used younger, non-diabetic people aged 18–50 [26].

### 2.3. Dietary Intervention

Fifty-one subjects without T2DM underwent two diets: HMD and HWD, in a 2-period randomized, crossover, double-blind design. Each diet was consumed for 4 weeks with an average 3-week washout period (usual diet) in between weight-stable diets. As described in detail elsewhere [24,25,29], total energy was matched in the HMD and the HWD. Dietary instructions were given to subjects by providing 8 different energy levels in accordance with BMI and gender, where recipe samples and daily meal plans were included. HMD consisted of 200–300 g red meat, ≥50 g processed meat, 4–6 servings of refined grains, 1–2 servings of vegetables, 1–2 servings of fruits, 200–300 g potatoes, 1 serving of jam or marmalade, 3–9 servings of oil or spread, or 3–4 servings of indulgence according to subjects’ weight. Meanwhile, HWD comprised 4 servings of low fat dairy products (including 2 servings of yoghurt), 3–4 servings of whole grain, 60–90 g unsalted nuts and either 70–150 g chicken or fish (or other seafood) or 150–225 g cooked legumes, 2–7 servings of oil or spread and 1 serving of jam or marmalade (based on subjects’ weight). Healthy cooking methods including steaming, boiling, stewing and poaching rather than deep-frying, grilling, and roasting were encouraged for the HWD. Food Works Professional Edition 8.0 (Xyris, Spring Hill, QLD, Australia) was used for analyses of food intake in 3-day weighed periods.

### 2.4. Assessments of Insulin Sensitivity 

The data for ISI from low-dose insulin and glucose infusion test (LDIGIT; *n* = 47) and homeostasis model assessment of insulin resistance (HOMA-IR; *n* = 49) was obtained from the previous study [16].

### 2.5. Biochemical Analysis

The data for glucose, insulin, TC, triglyceride (TG), HDL-C, hs-CRP, IL-6, CML (measured by ELISA), and total fluorescent AGEs (measured in a multi-mode microplate reader) has been previously published [24,25].

### 2.6. LC-MS/MS

#### 2.6.1. Materials

Lysine, CML, [2H2]-lysine (Lys), [2H2]-CML, [4H2]-CEL and [3H2]-MG-H1 were purchased from Iris Biotech (Adalbert-Zoellner-Str 1, Marktredwitz, Germany). Nonafluoropentanoic acid (NFPA; 396575), o-phthaldialdehyde (P0657), N-acetyl-L-cysteine (A7250), boric acid (B7901) and sodium hydroxide (55881) were obtained from Sigma-Aldrich (Sigma, St. Louis, MO, USA). Acetonitrile (BDH) was purchased from Prolabo. All of these reagents were of analytical grade. HPLC-grade acetonitrile was from BDH. D and l-Lysine-4,4,5,5-2H4·2 HCl (99% 2H4) were from CDN Isotopes.

#### 2.6.2. Blood Sample Preparation

Plasma was centrifuged at 4000 RPM at 4 °C for 10 min (Universal 32R, Hettich Zentrifugen, Germany). Protein-bound CML, CEL and MG-H1 levels in each sample were measured as previously described elsewhere [30,31,32]. Briefly, 100 μL of plasma samples were aliquoted for reduction with 20 μL 1 M sodium borohydride in 0.1 M sodium hydroxide. The glycated protein was precipitated with 20% of trichloroacetic acid (TCA) and, then, protein was hydrolyzed with 6 M hydrochloric acid (HCL) for 24 h at 100 ± 1 °C. Hydrolysates were spiked with [2H2]-lysine (Lys), [2H2]-CML, [4H2]-CEL and [3H2]-MG-H1 before solid-phase extraction (SPE). Sep-Pak (RP C18) columns were used for solid-phase extraction. The analyte of interest was eluted with 3 mL of 1% v/v trifluoroacetic acid (TFA) (in 20% v/v methanol) and dried under nitrogen and reconstituted in 1 ml of 20% v/v methanol prior to injection.

#### 2.6.3. LC-MS/MS

The LC-MS/MS method was used to determine protein-bound plasma concentrations of Nε-(carboxymethyl) lysine (CML), Nε-(1-carboxyethyl)lysine (CEL) and Nδ-(5-hydro-5-methyl-4-imidazolon-2-yl)-ornithine (MG-H1) using a Sciex QTRAP 6500+ liquid chromatography–mass spectrometer (Sciex, Framingham, MA, USA) with detection in ESI positive multiple reaction monitoring (MRM) mode. Derivatization of samples was performed on a reversed-phase C18 column (Phenomex Synergi hydro-4 μm particle size, 80 Å pore size, 150 × 4.6 mm (Phenomenex, Torrance, CA, USA)) with a linear gradient of 0.1% formic acid and 100% acetonitrile. Derivatized samples were injected (1 µL) at a flow rate of 0.4 ml/min over 6 min. Lys (147.4 > 83.9), CML (205.1 > 84), CEL (219.2 > 130), MG-H1 (229.2 > 116.1), [2H2]-Lys (151.2 > 87.9), [2H2]-CML (207.1 > 129.9), [4H2]-CEL (222.9 > 134.2) and [3H2]-MG-H1 (232.2 > 116.1) were used for MRM transitions. A standard calibration curve was prepared by a plot of analyte peak area divided by internal standard peak area (area ratio) against concentration (amount ratio). Results are presented as nmol/mL.

### 2.7. Statistical Analyses

Analyses were performed with SPSS V22 (IBM, Chicago, IL, USA). The Shapiro–Wilk test, Q-Q plots and histograms were used to investigate the normality of distribution. Log transformation was performed in the event that skewed variables were log transformed to approximate a normal distribution. Changes between two diets were tested with paired-samples *t*-tests or Wilcoxon signed-rank tests when variables were still skewed after log transformation. Unpaired *t*-tests, Mann–Whitney nonparametric tests and chi-square tests were conducted to contrast variables between groups. Pearson and Spearman correlation coefficients were used to determine the associations between variables. Data is presented as means ± SDs except for skewed data, which is presented as medians and interquartile ranges. Statistical significance was defined as *p* < 0.05. CEL levels were skewed and log transformation was performed prior to a paired *t*-test.

## 3. Results

### 3.1. The Effect of Two Diets on AGEs Levels

As we previously reported [24,25], a total of 51 subjects without T2DM (15 men and 36 women aged 35.1 ± 15.6 y; BMI, 27.7 ± 6.9 kg/m^2^) completed the intervention and 49 out of 52 subjects successfully completed LDIGIT, which was conducted at the end of each dietary intervention (HMD and HWD). A detailed explanation of why the other samples were not analyzed is shown in Figure 1. With regard to AGE measurement, plasma samples of 2 subjects among a total of 51 subjects were not available for AGE measurement. Therefore, plasma samples of 49 subjects (15 men and 34 women aged 35.5 ± 15.6 y; BMI, 27.7 ± 6.9 kg/m^2^) were measured by LC-MS/MS. Three values were excluded from the AGE analysis for technical reasons, so the final numbers are CML (*n* = 46), CEL (*n* = 48) and MG-H1 (*n* = 49).

The baseline characteristics of a total of 47 subjects who had both AGEs and ISI measured divided into 2 LDIGIT groups are shown in Table 1. The insulin-resistant group, as expected, had a higher BMI than the insulin-sensitive group, but this group was not prediabetic. BMI was unrelated to AGE levels overall.

The effects of the two diets on insulin sensitivity, biochemical characteristics and AGE levels are shown in Table 2. The levels of TG, total cholesterol and PAI-1 were significantly higher after HMD than after HWD. The levels of fasting glucose, fasting insulin, HOMA-IR, HDL-C and hs-CRP did not differ between the two diets.

The levels of CEL on the HMD were significantly higher compared with the HWD (1.367, 0.78 for HMD vs. 1.096, 0.65 nmol/mL for HWD; *p* = 0.01; *n* = 48). The levels of CML was not different between HMD and HWD (4.05 ± 1.44 for HMD vs. 3.85 ± 1.41 nmol/mL for HWD; *p* = 0.3). MG-H1 also did not differ between the two diets (6.47 ± 3.46 for HMD vs. 6.83 ± 2.83 nmol/mL for HWD; *p* = 0.25). When AGEs were separately analyzed in insulin-sensitive and insulin-resistant groups (Table 3), CEL on HMD was significantly higher than CEL on HWD in an insulin-resistant group (1.50, 0.85 for HMD vs. 1.13, 0.57 for HWD; *p* = 0.01; *n* =24), while no difference was observed in an insulin-sensitive group (1.32, 0.78 for HMD vs. 1.29, 1.059 for HWD; *p* = 0.7; *n* = 22).

No significant associations between AGEs (CML, CEL and MG-H1) and insulin sensitivity index (ISI _LDIGIT_ and HOMA-IR), fasting glucose, fasting insulin, lipids, PAI-1, hs-CRP or IL-6 on the two diets were seen.

### 3.2. Association between AGEs and Dietary Intake

Dietary intake examined by weighed food records during the dietary intervention period was previously reported [24]. No correlations between AGEs (CML, CEL and MG-H1) and dietary intake were observed.

## 4. Discussion

In this weight-stable 4-week randomized crossover design, a high-red-and-processed-meat diet and a low-fiber, medium glycemic index/glycemic load diet (HMD) led to a significant increase in plasma CEL concentrations compared with the intake of a diet high in dairy, whole grains, nuts and legumes and low in GI/glycemic load (HWD) in individuals without T2DM. However, HMD did not affect the plasma concentrations of CML and MG-H1 compared with HWD. Moreover, we found that plasma concentrations of CML, CEL and MG-H1 on either the HMD or HWD did not correlate with the insulin sensitivity index, fasting glucose, fasting insulin, lipids or inflammatory markers.

Western-type diets containing high AGEs are partially absorbed in the digestive tract leading to elevated total serum AGE levels with possible negative consequences on health [33]. CML is produced by both glycoxidation and lipid oxidation [34], but, in this study, we did not observe higher CML concentrations after HMD than after HWD.

In an observational study of healthy subjects, dietary AGE intake was associated with elevated blood concentrations of CML and MG as assessed by immunoassays independent of age or energy intake [35]. In contrast, the consumption of AGE-rich foods was not associated with blood CML concentrations in a second observational study of healthy subjects [36].

In an acute intervention of crossover design, 19 healthy overweight subjects consumed two test meals of high AGE and low AGE content. Concentrations of CML in plasma and urine as measured by LC-MS/MS were not significantly increased after high-AGE meals [37].

In the Cohort on Diabetes and Atherosclerosis Maastricht study (CODAM study) of 450 subjects with increased risk for T2DM and cardiovascular disease, consumption of dietary AGEs (CML, CEL and MG-H1)––as estimated with a dietary AGE database and a food frequency questionnaire––was positively associated with increased free plasma and urinary AGE concentrations, as assessed by UPLC-MS/MS. However, no significant association between dietary AGEs and protein-bound plasma AGEs was observed [38]. This differs from our finding of an increase in plasma protein-bound CEL from the high-red-and-processed-meat (HMD) diet—this may reflect the very high meat intake not normally seen in a free-living population.

In a double-blind randomized crossover study of 20 healthy overweight subjects, a high-AGE diet for 2 weeks did not alter serum concentrations of protein-bound CML, CEL and MG-H1 as measured by UPLC-MS/MS, compared with a low-AGE diet. However, this study showed significant lower urinary MG-H1 and CEL concentrations after a low-AGE diet. The two diets were matched for energy and macronutrient proportions from protein, fat, and carbohydrate [26]. Thus, most studies examining CML and MG-H1 did not see an increase in protein-bound AGE after a high-AGE diet, which is in line with our findings here.

In this present study, we found significantly higher plasma CEL concentrations after HMD than after HWD, but only in the insulin-resistant group. Our finding is supported by the observation that CEL is more likely to be contained in meat high in lipids, such as processed meat [39]. Interestingly, Kellow et al. 2017 [40] conducted a cross-sectional study to investigate if long-term habitual dietary patterns influence AGE accumulation in skin collagen. They found a positive association between meat consumption and skin AGE levels (r = 0.22; *p* < 0.05), and a negative association between cereal consumption and skin AGE levels (r = −0.21; *p* < 0.05) in 251 healthy overweight subjects (mean BMI 26 ± 6 kg/m^2^) aged 18–80 years. There were significant associations between skin AGE levels and age, anthropometric parameters, such as BMI (r = 0.23; *p* < 0.05), body weight (r = 0.24; *p* < 0.05) and waist circumference (r = 0.28; *p* < 0.01).

Courten et al 2016 [26] showed an effect of a high- and low-AGE diet on insulin sensitivity. The volunteers consumed higher amounts of CEL and CML on the high-AGE diet, so dietary CEL may contribute as much or more than CML to the effect on insulin sensitivity.

In our previous study, we found no differences in fluorescent AGEs or CML between HMD and HWD as assessed by immunoassays [25]. Consistent with this, protein-bound CML concentrations measured by LC-MS/MS in this study also did not differ between two diets.

A strength of this study is that we measured plasma protein-bound CML, CEL and MG-H1 using LC-MS/MS, which is an accurate, highly sensitive method [27,28,41]. Most intervention studies measured CML concentrations by immunological assays.

This study has several limitations. We did not measure free plasma AGE, which may provide more insight into the relationship between dietary AGE consumption and plasma AGE levels. We did not measure CML, CEL or MG-H1 concentrations in urine or fecal samples.

In conclusion, consumption of high red and processed meat and a low-fiber, medium-glycemic index/glycemic load diet in young individuals without T2DM significantly increased plasma CEL concentrations, but not plasma CML, and MG-H1 concentrations, compared with consumption of a diet high in dairy, whole grains, nuts and legumes and low in GI/glycemic load. Further investigations are needed to clarify the associations between dietary AGE consumption and AGE concentrations in blood (free and protein-bound), urine and feces, which will enable us to identify the role of dietary AGEs in human health. Future work should study the impact of dietary quality/patterns and cooking methods on AGEs, as well as food groups and macronutrients.

## Figures and Tables

**Figure 1 nutrients-12-01767-f001:**
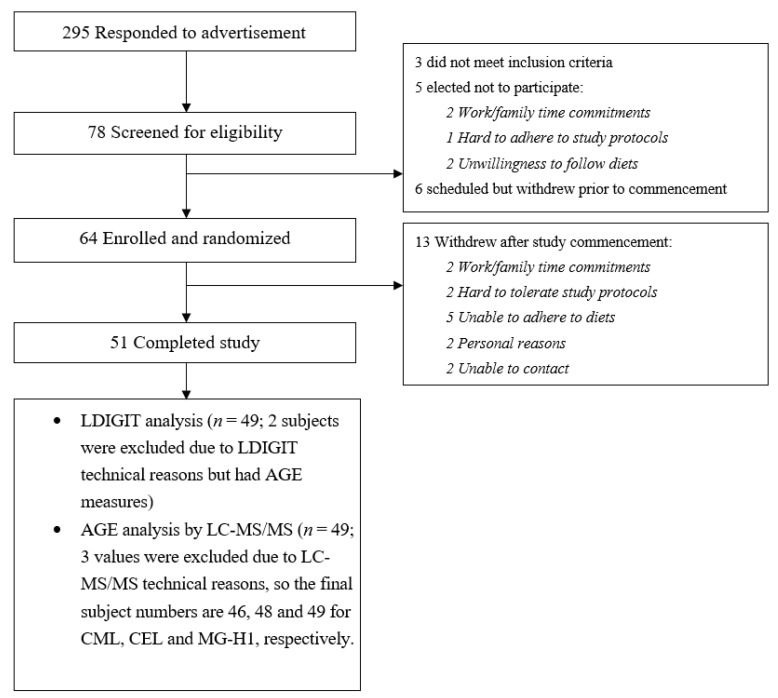
Flow diagram of subject recruitment and sample analysis explanation.

**Table 1 nutrients-12-01767-t001:** Baseline characteristics of subjects.

	All Subjects ^1^	Subjects Who Completed LDIGIT (*n* = 47)		
		Insulin-Sensitive Group (*n* = 23)	Insulin-Resistant Group (*n* = 24)	*p*
Sex (M/F)	15/34	9/14	6/18	0.3 ^†^
Age (y)	35.5 ± 15.6	36.0 ± 15.7	36.1 ± 16	0.98
NGT (n)	16	15	7	0.3
IFG/IGT (n)	33	8	17	
Baseline fasting glucose (mmol/L)	5.5 ± 0.7	5.3 ± 0.8	5.6 ± 0.6	0.09
Baseline 2 h glucose (mmol/L)	7.29 ± 1.6	7.05 ± 1.5	7.27 ± 1.6	0.2
HOMA-IR (HWD)	0.37, 0.43	0.3, 0.46	0.38, 0.47	0.13 ^‡^
HOMA-IR (HMD)	0.52, 0.67	0.22, 0.56	0.67, 0.86	0.01 ^‡^
LDIGIT _(120-150min)_ insulin (HWD pmol/L) ^2^		34.2, 28.9	120.7, 154	< 0.001 ^#^
LDIGIT _(120-150min)_ insulin (HMD pmol/L) ^2^		28.7, 20.1	152.8, 200	< 0.001^#^
Baseline weight (kg)	79.7 ± 21.36	70.5 ± 14.9	85.4 ± 22.1	0.01
BMI (kg/m^2^)	27.7 ± 6.9	24.6 ± 4.6	29.2 ± 5.9	0.005
Baseline SBP (mmHg)	112.5 ± 10.8 ^3^	110.7 ± 9.3 ^4^	114.7 ± 12.6 ^5^	0.3
Baseline DBP (mmHg)	70.6 ± 9.8 ^3^	69.0 ± 9.9 ^4^	72.6 ± 9.9 ^5^	0.2
Total Fat Mass (kg)	28.9 ± 15.7	20.7 ± 11.5	33.8± 13.3	0.001
Total Lean Mass (kg)	46.9 ± 11.5	46.0 ± 11.4	47.8 ± 12.3	0.6
Total Fat Mass (%)	36.5 ± 12.7	30.2 ± 13.1	40.7 ± 8.6	0.002

Groups were contrasted with an unpaired *t*-test. † *p* values were obtained by chi-square test. ‡ *p* values were obtained by Mann–Whitney nonparametric test. # *p* values were obtained by unpaired *t*-tests after log transformation. Values are means ± SDs except for HOMA-IR, LDIGIT _(120-150min)_ insulin and CEL, which are medians and interquartile ranges. The insulin-sensitive group and insulin-resistant group were defined a posteriori based on the insulin values of LDIGIT _120-150 min_, not from randomized groups: insulin-sensitive group <56 pmol/L with a median insulin of 33 pmol/L (*n* = 23) and insulin-resistant group >56 pmol/L with a median insulin of 122 pmol/L (*n* = 24). BMI, body mass index; CEL, carboxy ethyl lysine; CML, carboxymethyllysine; DBP, diastolic blood pressure; F, female; HOMA-IR, homeostasis model assessment of insulin resistance; IFG, impaired fasting glucose; IGT, impaired glucose tolerance; LDIGIT, low-dose insulin and glucose infusion test; M, male; MG-H1, methylglyoxal–hydroimidazalone; NGT, normal glucose tolerance; SBP, systolic blood pressure. ^1^
*n* = 49;^2^
*n* = 47, ^3^
*n* = 40; ^4^
*n* = 21; ^5^
*n* = 18.

**Table 2 nutrients-12-01767-t002:** Insulin sensitivity, lipids, inflammatory and fibrinolytic markers, and AGEs measured at the end of each dietary period in all participants (*n* = 49).

Metabolic Parameters	HMD	HWD	*p*
Fasting glucose (mmol/L)	5.27, 0.76	5.3, 0.55	0.9
Fasting insulin (µMmol/mL)	2.15, 1.97	2.17, 1.15	0.07
HOMA-IR	0.52, 0.67	0.37, 0.43	0.28
TG (mmol/L)	0.92, 0.74	0.77, 0.65	0.033
HDL-C (mmol/L)	1.39 ± 0.43	1.37 ± 0.42	0.4
Total cholesterol (mmol/L)	4.7 ± 0.96	4.6 ± 1.0	0.038
hs-CRP (mg/L)	0.79, 2.93	0.58, 2.24	0.79
IL-6 (pg/ml)	7.9, 11.2	7.7, 11	0.5
PAI-1 (ng/ml)	159, 83	121, 51	<0.001
CML (nmol/mL) ^1^	4.05 ± 1.44	3.85 ± 1.41	0.3
CEL (nmol/mL) ^2^	1.367, 0.78	1.096, 0.65	0.01
MG-H1 (nmol/mL)	6.47 ± 3.46	6.83 ± 2.83	0.25

*p* values for total cholesterol and HDL-C, CML and MG-H1 were determined by paired *t*-tests. *p* values for fasting glucose and TG and CEL were determined by paired *t*-test after log transformation. *p* values for fasting insulin, HOMA-IR, hs-CRP, PAI-1, IL-6 and CEL were obtained by Wilcoxon signed-rank nonparametric test. Values (*n* = 49) are expressed as medians and interquartile ranges except for HDL-C, total cholesterol, CML and MG-H1, which are presented as means ± SDs. CEL, carboxy ethyl lysine; CML, carboxymethyllysine; HMD, a diet high in red and processed meat and refined grains; HWD, a diet high in whole grain, nuts, dairy and legumes; HOMA-IR, homeostasis model assessment of insulin resistance; HDL-C, high-density lipoprotein cholesterol; hs-CRP, high-sensitivity C-reactive protein; IL-6, interleukin 6; MG-H1, methylglyoxal–hydroimidazalone; PAI-1, plasminogen activator inhibitor type 1; TG, triglyceride; ^1^
*n* = 46; ^2^
*n*= 48.

**Table 3 nutrients-12-01767-t003:** Advanced glycation end product levels in diets separately analysed by insulin sensitive/resistant groups.

	Insulin-Sensitive Group				Insulin-Resistant Group			
	HMD	HWD	N	*p*	HMD	HWD	N	*p*
CML (nmol/mL)	4.34 ± 1.55	4.25 ± 1.4	21	0.79	3.91 ± 1.34	3.59 ± 1.37	23	0.2
CEL (nmol/mL)	1.32, 0.78	1.29, 1.059	22	0.7	1.50, 0.85	1.13, 0.57	24	0.01
MG-H1 (nmol/mL)	6.09 ± 2.37	6.52 ±2.18	23	0.36	7.01 ± 4.32	7.05 ± 3.46	24	0.9

*p* values for CML and MG-H1 were determined by paired *t*-tests. *p* values for CEL were determined by paired *t*-tests after log transformation. Values are means ± SDs except for CEL, which are medians and interquartile ranges CEL, carboxy ethyl lysine; CML, carboxymethyllysine; HMD, a diet high in red and processed meat and refined grains; HWD, a diet high in whole grain, nuts, dairy and legumes; MG-H1, methylglyoxal–hydroimidazalone.

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
