# Peer review of "Differential Effects of Dietary Patterns on Advanced Glycation end Products: A Randomized Crossover Study"

_nutrients, 2020, doi:10.3390/nu12061767_

Round 1
Reviewer 1 Report
Introduction: Some paragraphs can probably be moved to the discussion. A rationale for why individuals without T2DM is lacking, or maybe it's just not fully understood yet? Please mention the reason why the study population was chosen. If AGE increases with age, was there a rationale for choosing participants >18 years old, rather than older adults? What were the previous age ranges from the other studies here mentioned?
Results: Not clear the n. If 51 subjects completed the study but only samples of 49 individuals were analyzed and then only 47 had baseline values? This is a bit confusing. Please make changes to 47 subjects if that is what is being analyzed throughout the manuscript, and describe of make a flow diagram to mention the reason why the other samples were not analyzed (technical difficulties?). The results make more sense when the same n of subjects is being used throughout all the analysis. If the analysis is done with the 47 participants only, does it changes the results?
- The insulin sensitivity test (LDIGIT) was performed only at baseline? After the intervention, if the groups are divided again according to what was done in the baseline, into insulin-sensitive and insulin-resistant groups, are there any significant changes that would explain the role of AGE diet in these metabolic parameters? Suggestion: add as supplementary figure.
- Design of the table are different. Please fix that to assure that formatting of both tables are the same.
- BMI seems to be significant different amongst groups. Did the researchers do any analysis or correlation to BMI or separated the subjects into the different BMI categories to see if AGE was related to different BMI?
- Is this population pre-diabetic?
Discussion: Studies don't seem to find a significance in plasma CEL, but it is not the case for this manuscript. Please describe in more details besides. the Kellow et al 2017 paper. Are there any in vivo or in vitro studies that would help describe the importance or role of high CEL and why a HWD diet would be best recommended?
- missing discussion for other findings OR explanation to the baseline results for this population. As a reader, I want to know why the authors chose to represent the baseline data into those groups, but didn't do the same for after the intervention?
- TG, TC, PAI-1 are significant after intervention, was this previously observed? That might be best to move the paragraph from the introductions to the discussion.
Author Response
Comments and Suggestions for Authors
Introduction: Some paragraphs can probably be moved to the discussion.
We have shortened much of the detail on the previous study in the introduction.
A rationale for why individuals without T2DM is lacking, or maybe it's just not fully understood yet? Please mention the reason why the study population was chosen. If AGE increases with age, was there a rationale for choosing participants > 18 years old, rather than older adults? What were the previous age ranges from the other studies here mentioned? Dietary source
This study examined the influence of diet on protein-bound AGEs. In older people and people with T2DM, the levels of protein bound AGEs are high and it would be difficult to see any influence of short-term dietary changes on protein-bound AGEs. Thus, we hoped with a population with a mean age of 35 years we would be more likely to see dietary influences. The only previous study examining dietary AGEs and insulin sensitivity used younger, non-diabetic people aged 18-50 [26].
We have added some of this detail to the methods.
In lines 53-58,
It is uncertain whether dietary AGEs can play a role in the etiology of type 2 diabetes mellitus (T2DM), or cardiovascular disease (CVD) whether only endogenously generated AGEs are related to these diseases. However, high red meat consumption (especially processed meat) as part of a low antioxidant, low fibre highly available starch standard western diet can increase dietary AGEs.[21]. The effect of dietary AGEs on disease risk markers can differ according to their health status (healthy, with or without T2DM or with CVD) [2,22,23].
Results:
Not clear the n. If 51 subjects completed the study but only samples of 49 individuals were analyzed and then only 47 had baseline values? This is a bit confusing. Please make changes to 47 subjects if that is what is being analyzed throughout the manuscript, and describe of make a flow diagram to mention the reason why the other samples were not analyzed (technical difficulties?).
Thank you. It has been addressed in line 198 and a diagram in Figure 1 has been created on page 6.
-Detailed explanation on why the other samples were not analyzed was shown in Figure 1 and in the results.
2 subjects were excluded from the insulin sensitivity analysis but had AGE measures. 3 values were excluded from the AGE analysis for technical reasons so the final numbers are
CML (n=46), CEL (n=48), MG-H1 (n =49)
The results make more sense when the same n of subjects is being used throughout all the analysis. If the analysis is done with the 47 participants only, does it change the results?
Thank you. We have included all available values to maximize power.
- The insulin sensitivity test (LDIGIT) was performed only at baseline? After the intervention, if the groups are divided again according to what was done in the baseline, into insulin-sensitive and insulin-resistant groups, are there any significant changes that would explain the role of AGE diet in these metabolic parameters? Suggestion: add as supplementary figure. Thank you. As we previously reported in the publication (Kim, Y.; Keogh, J.B.; Clifton, P.M. Consumption of red and processed meat and refined grains for 4 weeks decreases insulin sensitivity in insulin-resistant adults: A randomized crossover study. Metabolism: clinical and experimental 2017, 68, 173-183), LDIGIT was performed at the end of each diet. LDIGIT was not conducted at baseline. LDIGIT was not performed at baseline or after washout so the primary endpoint of the publication was a comparison of insulin sensitivity at the end of the two diets. When AGEs were separately analyzed in insulin sensitive and insulin resistant group (Table 3), CEL on HMD was significantly higher than CEL on HWD in an insulin resistant group (1.50, 0.85 for HMD vs 1.13, 0.57 for HWD; P = 0.01; n =24), while no difference was observed in insulin sensitive group (1.32, 0.78 for HMD vs 1.29, 1.059 for HWD; P 0.7; n = 22).
- The following results are added in lines 212-215.
- We have added AGE levels to the diet/insulin sensitivity Table 3.
- Design of the table are different. Please fix that to assure that formatting of both tables is the same.
- Thank you. The formatting of tables has been fixed.
- BMI seems to be significant different amongst groups. Did the researchers do any analysis or correlation to BMI or separated the subjects into the different BMI categories to see if AGE was related to different BMI? Is this population pre-diabetic?Thank you. It has been addressed in lines 201-203. Discussion:
- Yes, “The Insulin resistant group as expected had a higher BMI than the insulin sensitive group but this group was not prediabetic. BMI was unrelated to AGE levels overall”.
- Studies don't seem to find a significance in plasma CEL, but it is not the case for this manuscript. Please describe in more details besides. the Kellow et al 2017 paper. Thank you. The Kellow et al 2017 paper has been described in detail in lines 338-344 Kellow et al 2017 [39] conducted a cross -sectional study to investigate if long-term habitual dietary patterns influence AGE accumulation in skin collagen. They found a positive association between meat consumption and skin AGEs levels (r = 0.22; P < 0.05), and a negative association between cereal consumption and skin AGE levels (r = -0.21; P < 0.05) in 251 healthy overweight subjects (Mean BMI 26 ± 6 kg/m2) aged 18-80 years. There was a significant association between skin AGE levels and age, anthropometric parameters such as BMI (r = 0.23; P < 0.05), body weight (r = 0.24; P < 0.05) and waist circumference (r = 0.28; P < 0.01).
Are there any in vivo or in vitro studies that would help describe the importance or role of high CEL and why a HWD diet would be best recommended?
Thank you. It has been addressed in lines 345-347
Courten et al 2016 [26] showed an effect of a high and low AGE diet on insulin sensitivity. The volunteers consumed higher amounts of CEL and CML on the high AGE diet so dietary CEL may contribute as much or more than CML to the effect on insulin sensitivity.
- missing discussion for other findings OR explanation to the baseline results for this population. As a reader, I want to know why the authors chose to represent the baseline data into those groups, but didn't do the same for after the intervention?Sorry we have corrected that omission and added the data to Table 3. We explored the relationship between insulin sensitivity and AGE and found no effects and mentioned that in the results.
- TG, TC, PAI-1 are significant after intervention, was this previously observed? That might be best to move the paragraph from the introductions to the discussion.
We have omitted this data from the introduction along with other details from the previous paper.
Reviewer 2 Report
General Comments
In this work, the authors have performed follow-up analysis on a prior conducted randomized controlled trial that compared the effects of a processed foods diets (high in red meat and refined grains) versus what would be considered a “healthier” unprocessed diet (high in dairy/chicken/nuts/whole grains) on inflammatory markers, lipoprotein profiles, insulin sensitivity and AGEs (CML and fluorescent AGEs specifically). In the present work, the authors extend this work by again studying CML with the addition of CEL and MG-H1. The authors also use LC/MS/MS based assays in the current works as opposed to ELISAs in their previous work. The authors discuss an important topic, namely the formation of AGE’s in a standard Western-diet (rich in processed foods, low in fruits/vegetables and high in red/processed meat). This is a timely analysis given the concern about such diets on AGE formation and their implicated role in the development of metabolic disease.
I have several general comments, that I would feel need to be addressed.
- The authors should at least mention in the intro that CML and fluorescent AGEs were already measured in prior analysis of this study, which would put the mentioning of using LC/MS/MS into better perspective. The reader is now left to find out you studied CML in the same data set near the end of the manuscript.
- The authors should also discuss why LC/MS/MS may be a more reliable method to study AGEs, as otherwise it seems like you just report the same CML findings twice. The discrepancy in findings between the current and present analysis should also be mentioned. In particular, p=0.07 in previous study, trend towards higher AGE as assessed by CML in HWD diet whereas you found lower CEL concentrations in the present work. Why would different AGE markers give you different results, this is important to discuss (is it methodological or physiological).
- The authors should discuss the impact of red meat and food processing methods (baking, grilling etc.) in the context of dietary patterns. There is plenty of evidence to suggest that the formation of AGEs is pretty much abolished when red meat is consumed with plenty of unprocessed plant foods (spices, herbs, fruits and vegetables) See (Pierre et al., 2003;Vulcain et al., 2005;Gorelik et al., 2008;Hur et al., 2009;Li et al., 2010;Van Hecke et al., 2017). Yes, high red meat can be an issue but its effects are likely perpetuated when consumed as part of a Standard Western Diet (see White and Collinson, 2013)
I offer the following specific comments:
Lines 29-30: These sentences can be combined: e.g., “AGEs are associated with accelerated aging and may be involved in the development of degenerative diseases”….
Line 33: Delete the words “a heterogenous group of compounds”. You describe AGEs right after and these words do not add any additional information
Line 43: need comma after “product”
Line 46: Under dry conditions instead of in dry conditions
Line 58: add the word “also” between diets and reduced.
Line 63: need the word “in” after “Moreover, “.
Lines 52:62: It is sufficient to highlight the key findings of the meta-analysis and discuss these on broader term rather than providing as much detail. The reader can look that up if desired.
Line 205: As you have no baseline data, you cannot say that the CEL increased in the HMD group when compared to the HWD groups. This needs to be changed to higher.
Line 282: delete the word “of”.
Line 296: too many commas. Please revise for readability.
Line 302: this sentence needs to be revised for readability; the end of the sentence can for instance be changed to “which is in line with our findings here”.
Line 314: This statement needs a reference (LC being more reliable than IA).
Line 324: The conclusion can be improved in my opinion. It should be mentioned that future work should study the impact of dietary quality/patterns and how they modulate these associations, which is in line with my comment earlier.
References.
Gorelik, S., Ligumsky, M., Kohen, R., and Kanner, J. (2008). The stomach as a "bioreactor": when red meat meets red wine. J Agric Food Chem 56, 5002-5007.10.1021/jf703700d
Hur, S.J., Lim, B.O., Park, G.B., and Joo, S.T. (2009). Effects of Various Fiber Additions on Lipid Digestion during In Vitro Digestion of Beef Patties. Journal of Food Science 74, C653-C657.10.1111/j.1750-3841.2009.01344.x
Li, Z., Henning, S.M., Zhang, Y., Zerlin, A., Li, L., Gao, K., Lee, R.-P., Karp, H., Thames, G., Bowerman, S., and Heber, D. (2010). Antioxidant-rich spice added to hamburger meat during cooking results in reduced meat, plasma, and urine malondialdehyde concentrations. The American Journal of Clinical Nutrition 91, 1180-1184.10.3945/ajcn.2009.28526
Pierre, F., Taché, S., Petit, C.R., Van Der Meer, R., and Corpet, D.E. (2003). Meat and cancer: haemoglobin and haemin in a low-calcium diet promote colorectal carcinogenesis at the aberrant crypt stage in rats. Carcinogenesis 24, 1683-1690
Van Hecke, T., Ho, P., Goethals, S., and De Smet, S. (2017). The potential of herbs and spices to reduce lipid oxidation during heating and gastrointestinal digestion of a beef product. Food research international 102, 785-792
Vulcain, E., Goupy, P., Caris-Veyrat, C., and Dangles, O. (2005). Inhibition of the metmyoglobin-induced peroxidation of linoleic acid by dietary antioxidants: Action in the aqueous vs. lipid phase. Free Radical Research 39, 547-563.10.1080/10715760500073865
White, D.L., and Collinson, A. (2013). Red Meat, Dietary Heme Iron, and Risk of Type 2 Diabetes: The Involvement of Advanced Lipoxidation Endproducts. Advances in Nutrition 4, 403-411.10.3945/an.113.003681
Author Response
General Comments
In this work, the authors have performed follow-up analysis on a prior conducted randomized controlled trial that compared the effects of a processed foods diets (high in red meat and refined grains) versus what would be considered a “healthier” unprocessed diet (high in dairy/chicken/nuts/whole grains) on inflammatory markers, lipoprotein profiles, insulin sensitivity and AGEs (CML and fluorescent AGEs specifically). In the present work, the authors extend this work by again studying CML with the addition of CEL and MG-H1. The authors also use LC/MS/MS based assays in the current works as opposed to ELISAs in their previous work. The authors discuss an important topic, namely the formation of AGE’s in a standard Western-diet (rich in processed foods, low in fruits/vegetables and high in red/processed meat). This is a timely analysis given the concern about such diets on AGE formation and their implicated role in the development of metabolic disease.
I have several general comments, that I would feel need to be addressed.
- The authors should at least mention in the intro that CML and fluorescent AGEs were already measured in prior analysis of this study, which would put the mentioning of using LC/MS/MS into better perspective. The reader is now left to find out you studied CML in the same data set near the end of the manuscript.Thank you. It has been addressed in lines 73-75 and 76-79.We have already measured CML by immunoassay and fluorescent AGEs in our previous publication [25].
- This present study aimed to investigate the association between dietary patterns and plasma AGEs (CML, CEL and MG-H1) using LC-MS/MS which is a far more specific and sensitive method as the immunoassay may suffer from steric hindrance and interference from anti-AGE antibodies [27,28].
- The authors should also discuss why LC/MS/MS may be a more reliable method to study AGEs, as otherwise it seems like you just report the same CML findings twice. Thank you. It has been addressed in lines 74-75 with two references However, the method of liquid chromatography-tandem mass spectrometry (LC-MS/MS) may be a more reliable method to study AGEs [27,28].27. Dorrian, C.A.; Cathcart, S.; Clausen, J.; Shapiro, D.; Dominiczak, M.H. Factors in human serum interfere with the measurement of advanced glycation endproducts. Cellular and molecular biology (Noisy-le-Grand, France) 1998, 44, 1069-1079.28.Shibayama, R.; Araki, N.; Nagai, R.; Horiuchi, S. Autoantibody against N(epsilon)-(carboxymethyl)lysine: an advanced glycation end product of the Maillard reaction. Diabetes 1999, 48, 1842-1849, doi:10.2337/diabetes.48.9.1842.Th discrepancy in findings between the current and present analysis should also be mentioned. In particular, p=0.07 in previous study, trend towards higher AGE as assessed by CML in HWD diet whereas you found lower CEL concentrations in the present work. Why would different AGE markers give you different results, this is important to discuss (is it methodological or physiological).Although our previous paper showed no significant difference between the diets in the level of plasma CML using an immuno-assay there were higher levels on the HWD, the opposite to what is seen using the LC/MS/MS method, although again the difference was not significant. We believe the differences are due to assay issues.
- We already measured CML and fluorescent AGEs as assessed by either fluorescence or immunoassays in prior analysis of our study [24].
- The authors should discuss the impact of red meat and food processing methods (baking, grilling etc.) in the context of dietary patterns. There is plenty of evidence to suggest that the formation of AGEs is pretty much abolished when red meat is consumed with plenty of unprocessed plant foods (spices, herbs, fruits and vegetables) See (Pierre et al., 2003;Vulcain et al., 2005;Gorelik et al., 2008;Hur et al., 2009;Li et al., 2010;Van Hecke et al., 2017). Yes, high red meat can be an issue but its effects are likely perpetuated when consumed as part of a Standard Western Diet (see White and Collinson, 2013)
Thank you. It has been addressed in lines 52-53 and in lines 55-57.
The formation of AGEs is reduced when red meat is consumed with abundant unprocessed plant foods (spices, herbs, fruits and vegetables) [15-20].
However, high red meat consumption (especially processed meat) as part of a low antioxidant, low fibre highly available starch standard western diet can increase dietary AGEs.
I offer the following specific comments:
Lines 29-30: These sentences can be combined: e.g., “AGEs are associated with accelerated aging and may be involved in the development of degenerative diseases”.
Thank you. It has been revised as follows;
The accumulation of advanced glycation end products (AGEs) is associated with accelerated aging [1] and may be involved in the development of degenerative disease…
Line 33: Delete the words “a heterogenous group of compounds”. You describe AGEs right after and these words do not add any additional information
Thank you. It has been deleted.
Line 43: need comma after “product”
Thank you. Comma has been inserted after “product”.
Line 46: Under dry conditions instead of in dry conditions
Thank you. It has been changed from “in” to “under”.
Line 58: add the word “also” between diets and reduced.
Thank you. Sentences have been revised.
Line 63: need the word “in” after “Moreover, “.
Thank you. Sentences have been revised.
Lines 52:62: It is sufficient to highlight the key findings of the meta-analysis and discuss these on broader term rather than providing as much detail. The reader can look that up if desired.
Thank you. It has been addressed in lines 59-63.
A meta-analysis of 17 randomized controlled trials (RCTs) comparing low AGE diets with high AGE diets indicated that low AGE diets could reduce risk markers of cardiometabolic disease such as insulin, LDL cholesterol, CRP and adhesion molecules and inflammatory markers [22]. A second smaller meta-analysis found similar effects and noted a high AGE diet increased circulating AGEs in most but not all studies (as assessed using ELISA) [2].
Line 205: As you have no baseline data, you cannot say that the CEL increased in the HMD group when compared to the HWD groups. This needs to be changed to higher.
Thank you. It has been revised.
The levels of CEL on the HMD were significantly higher compared with the HWD.
Line 282: delete the word “of”.
Thank you. It has been deleted.
Line 296: too many commas. Please revise for readability.
Thank you. It has been revised.
In a double-blind randomized crossover study of 20 healthy overweight subjects,
Line 302: this sentence needs to be revised for readability; the end of the sentence can for instance be changed to “which is in line with our findings here”.
Thank you. It has been revised.
Thus, most studies examining protein bound CML and MG-H1 did not see an increase in serum content after a high AGE diet, which is in line with our findings here.
Line 314: This statement needs a reference (LC being more reliable than IA).
Thank you. The references [27,28,41] have been cited.
- Dorrian, C.A.; Cathcart, S.; Clausen, J.; Shapiro, D.; Dominiczak, M.H. Factors in human serum interfere with the measurement of advanced glycation endproducts. Cellular and molecular biology (Noisy-le-Grand, France) 1998, 44, 1069-1079.
- Shibayama, R.; Araki, N.; Nagai, R.; Horiuchi, S. Autoantibody against N(epsilon)-(carboxymethyl)lysine: an advanced glycation end product of the Maillard reaction. Diabetes 1999, 48, 1842-1849, doi:10.2337/diabetes.48.9.1842.
- Teerlink, T.; Barto, R.; Ten Brink, H.J.; Schalkwijk, C.G. Measurement of Nepsilon-(carboxymethyl)lysine and Nepsilon-(carboxyethyl)lysine in human plasma protein by stable-isotope-dilution tandem mass spectrometry. Clinical chemistry 2004, 50, 1222-1228, doi:10.1373/clinchem.2004.031286.
Line 324: The conclusion can be improved in my opinion. It should be mentioned that future work should study the impact of dietary quality/patterns and how they modulate these associations, which is in line with my comment earlier.
Thank you. The suggested sentence has been added in conclusion.
The future work should study the impact of dietary quality/patterns and cooking methods on AGEs as well as food groups and macronutrients.
Round 2
Reviewer 1 Report
The authors have addressed the previous comments in an excellent manner. The manuscript is nicely done, and it slows much better. I highly recommend this manuscript for publication, and appreciate the authors efforts in addressing my suggestions and comments.
Author Response
The authors have addressed the previous comments in an excellent manner. The manuscript is nicely done, and it slows much better. I highly recommend this manuscript for publication, and appreciate the authors efforts in addressing my suggestions and comments.
Thank you.
Reviewer 2 Report
The authors have done an excellent job in improving the manuscript. I only have one minor comment:
"The formation of AGEs is reduced when red meat is consumed with abundant unprocessed plant foods (spices, herbs, fruits and vegetables) [15-20]."
Several of these studies are in vitro and in vivo in animals and not based on clinical studies, thus the sentence should be revised to a more cautious tone. I suggest something like:
"The formation of AGEs can potentially be reduced when red meat is consumed with abundant unprocessed plant foods (spices, herbs, fruits and vegetables) [15-20].
Author Response
The authors have done an excellent job in improving the manuscript. I only have one minor comment:
"The formation of AGEs is reduced when red meat is consumed with abundant unprocessed plant foods (spices, herbs, fruits and vegetables) [15-20]."
Several of these studies are in vitro and in vivo in animals and not based on clinical studies, thus the sentence should be revised to a more cautious tone. I suggest something like:
"The formation of AGEs can potentially be reduced when red meat is consumed with abundant unprocessed plant foods (spices, herbs, fruits and vegetables) [15-20].
Thank you. The sentence in blue has been revised as the reviewer suggests.
In lines 52-53,
“The formation of AGEs can potentially be reduced when red meat is consumed with abundant unprocessed plant foods (spices, herbs, fruits and vegetables) [15-20].”